# Infectious Bronchitis Virus: A Comprehensive Multilocus Genomic Analysis to Compare DMV/1639 and QX Strains

**DOI:** 10.3390/v14091998

**Published:** 2022-09-09

**Authors:** Ana P. da Silva, Rachel Jude, Rodrigo A. Gallardo

**Affiliations:** Department of Population Health and Reproduction, School of Veterinary Medicine, University of California, 1089 Veterinary Medicine Drive, 4009 VM3B, Davis, CA 95616, USA

**Keywords:** infectious bronchitis, viral evolution, whole genome sequencing, DMV, QX

## Abstract

Infectious bronchitis virus (IBV) is a highly variable RNA virus that affects chickens worldwide. Due to its inherited tendency to suffer point mutations and recombination events during viral replication, emergent IBV strains have been linked to nephropathogenic and reproductive disease that are more severe than typical respiratory disease, leading, in some cases, to mortality, severe production losses, and/or unsuccessful vaccination. QX and DMV/1639 strains are examples of the above-mentioned IBV evolutionary pathway and clinical outcome. In this study, our purpose was to systematically compare whole genomes of QX and DMV strains looking at each IBV gene individually. Phylogenetic analyses and amino acid site searches were performed in datasets obtained from GenBank accounting for all IBV genes and using our own relevant sequences as a basis. The QX dataset studied is more genetically diverse than the DMV dataset, partially due to the greater epidemiological diversity within the five QX strains used as a basis compared to the four DMV strains from our study. Historically, QX strains have emerged and spread earlier than DMV strains in Europe and Asia. Consequently, there are more QX sequences deposited in GenBank than DMV strains, assisting in the identification of a larger pool of QX strains. It is likely that a similar evolutionary pattern will be observed among DMV strains as they develop and spread in North America.

## 1. Introduction

Infectious bronchitis virus (IBV) is an avian coronavirus that affects predominantly the upper respiratory tract of chickens [1]. The IBV viral genome is composed of a single stranded, positive sense RNA molecule of nearly 30 kbp. In light of its genomic nature, IBV is prone to constant evolutionary changes, either by point mutations that are not proofread by its RNA-dependent RNA polymerase, or by recombination between subgenomic RNA molecules during viral replication [2]. For this reason, strains with modified pathogenicity and tissue tropism have emerged and caused additional losses to the poultry industry worldwide since the early 1950s [3]. Pragmatically, IBV strains that induce nephropathogenic and reproductive disease have been associated with more severe disease than the typical respiratory strains, with increased mortality and reports of unsuccessful vaccination [4].

The IBV QX strains emerged in China in the mid-1990s and were initially associated with proventriculitis [5,6]. Subsequently, variant strains spread throughout Asia and Europe, where reports of nephropathogenic disease associated with QX rapidly increased [7,8,9,10,11]. In the 2010s, QX was proposed as a possible causative agent of false layer syndrome (FLS) [12,13,14], a disorder in which sexually mature chickens ovulate normally but are unable to lay eggs [1]. This syndrome is thought to be a consequence of very early infections with IBV, hampering the oviductal development and consequently leading to the narrowing of the oviduct with possible cystic formations in the reproductive tract [3,15,16,17].

On a similar note, outbreaks of nephropathogenic disease associated with IBV in broiler chickens began in the Delaware, Maryland, and Virginia (Delmarva) peninsula in 2011, originating the so-called DMV/1639/11 genotype [18]. Almost a decade later, the DMV/1639 variants have been linked to FLS cases in Eastern Canada and the United States [19,20,21].

QX and DMV variant strains have shown a similar trajectory in terms of clinical signs and lesions since their emergence, initially causing high mortality due to a nephropathogenic disease and subsequently affecting the reproductive tract of chickens. However, genomic analyses do not show similarities between QX and DMV/1639. Using the current IBV classification method based on the S1 gene, these strains belong to distantly related lineages, with DMV/1639 strains belonging to lineage 17 and QX strains belonging to lineage 19 of genotype I [22]. The objective of this study was to compare whole genomes of DMV/1639 and QX strains to look for similarities and divergences in each IBV gene independently.

## 2. Materials and Methods

### 2.1. Isolates

Virus isolation was performed from tissues of affected birds (tracheas, kidneys, cecal tonsils, or intestines), homogenized in PBS and inoculated in the allantoic cavity of 9- to 11-day-old specific pathogen free (SPF) embryonating eggs (Charles River Laboratories, Willimantic, CT, USA), as previously described [23]. From 3 to 7 days post-inoculation, embryos presenting lesions (stunting, curling, and urate deposits in kidneys) were selected and allantoic fluid was harvested for further processing. All virus strains induced clinical signs in embryos in the very first passage, and the allantoic fluid from affected eggs/embryos were used for further experimental analyses and testing.

Four isolates used in this experiment were previously classified as DMV/1639-like (genotype I, lineage 17) based on the IBV S1 gene [22]. Isolate FLS/AZ/17 (OM525798) has been previously used in live bird challenge experiments to assess its potential to induce FLS compared to M41 [24]. The DMV25, DMV26 and DMV28 isolates were retrieved from FLS-affected layer chicken flocks in the Ohio/Indiana border that did not reach peak of production and presented with cystic ovaries upon postmortem evaluation. The GenBank accession numbers for the DMV/1639 whole genomes are OM525798-OM5257801.

The five QX isolates used in this project are from Eurasia (China, France, Greece, Hungary, and Slovakia) and were previously typed as genotype I, lineage 19 [22]. All isolates, except the one from France, originated from broiler chickens presenting with kidney lesions and/or respiratory disease. The French isolate was retrieved from 10-day-old layer birds with respiratory signs. The pathogenicity of these QX strains has been previously assessed in challenge experiments [12]. The GenBank accession numbers for the QX whole genomes are OM525802-OM5258806.

### 2.2. Library Preparation and Sequencing

Total RNA was extracted from allantoic fluids using a combination of TRIzol LS reagent (Thermo Fisher Scientific, Carlsbad, CA, USA) and Direct-zol MiniPrep Plus kit (Zymo Research, Orange, CA, USA), as previously described [25]. RNA integrity was checked with a bioanalyzer using the Agilent RNA 6000 Nano Kit (Agilent, Santa Clara, CA, USA). Oligo dT-based RNA selection was performed with the NEBNext Poly(A) mRNA Magnetic Isolation Module Kit (New England Biolabs, Ipswich, MA, USA). The reverse transcription of poly A-tailed RNA and cDNA library construction was performed using the NEBNext Ultra II RNA Library Prep kit for Illumina (New England Biolabs).

The libraries were sequenced using the Illumina HiSeq 4000 platform by pair-end 150-bp reads (Illumina, San Diego, CA, USA). Low quality sequences were removed, and adaptors were trimmed using the BBDuk plugin in Geneious Prime 2022.0.2 (Biomatters Ltd., Auckland, New Zealand). The trimmed reads were mapped to a reference IBV M41 genome (GenBank accession number AY851295). A second round of reference-based assembly was carried out using the consensus sequence obtained in the first round as a reference. For accuracy verification, de novo assembly was performed using the SPAdes assembler. The equivalent assemblies shared 100% nucleotide identity.

### 2.3. Infectious Bronchitis Virus Genotyping and Classification

A total of 173 sequences belonging to all 8 IBV genotypes and 37 lineages [22,26,27,28,29] were aligned with the 9 sequences from this study using the MAFFT plugin [30] in Geneious Prime 2022.1.1. A phylogenetic tree based on the S1 gene was constructed to categorize the IBV strains from this study into the current genotypic classification [22]. The maximum likelihood method based on the GTRGAMMAI model with 1000 bootstraps was used to build the phylogenetic tree with the RaxML plugin [31] in Geneious Prime 2022.1.1. The GenBank accession numbers of the sequences are listed in Figure 1.

### 2.4. Multilocus Genomic Analyses

The nine IBV sequences were divided into IBV lineages 17 (DMV/1639, n = 4) and 19 (QX, n = 5) from genotype I. BLAST searches were performed for each isolate by individual viral genes, namely: ORF1a, ORF1ab, spike (S), 3a, 3b, envelope (E), membrane (M), 5a, 5b, and nucleocapsid (N). For each BLAST search (nine IBV sequences and 10 genes), the top 5 nucleotide identity hits were selected, representing at least 5 sequences per search. Only whole genomes were selected for setting up the datasets. The sequences obtained in GenBank were sorted into DMV and QX datasets and repeated sequences were removed. Once the two datasets were created, alignments and phylogenetic trees were created separately by gene for DMV and QX as described in Section 2.3.

### 2.5. Prediction of Antigenic Regions

To predict potential antigenic sites within each protein of the viral genome, the EMBOSS Antigenic tool was used (https://bioinformatics.nl/cgi-bin/emboss/antigenic (accessed on 4 July 2022). This tool predicts potential antigenic regions on the surface of a protein by detecting hydrophobic regions containing cysteine, leucine, and valine, which are more likely to be part of antigenic sites than other amino acids [32]. The number of antigenic sites was measured from individual genes of QX and DMV datasets determined in Section 2.4. The number of antigenic sites was divided by the number of amino acid residues per gene and multiplied by 100 for a percentage (Appendix A). The average and standard deviation of these ratios were taken for each sequence of each dataset (QX and DMV). The higher the average, the greater the chances of having antigenic sites on the protein. The higher the standard deviation, the greater the variability within the dataset.

### 2.6. Statistical Analyses

All phylogenetic analyses were performed using the maximum likelihood method as described in Section 2.3.

The antigenic region ratios for each gene were compared using the nonparametric Kruskal–Wallis test followed by Dunn’s multiple comparisons test in Prism 9 (GraphPad, La Jolla, CA, USA). Two-sample one-tailed F tests were used to measure how spread out the antigenic region ratio datapoints were in each gene. Since this is a one-tailed test, the greater variance between QX and DMV/1639 was used as the numerator. The descriptive statistics were calculated using Microsoft Excel (v. 16.6) with the Analysis ToolPak add-on. When F is greater than F_critical_, the null hypothesis that the variances of the two datasets are equal is rejected (*p* < 0.05).

## 3. Results

### 3.1. Infectious Bronchitis Virus Genotyping and Classification

Based on the S1 gene [22], four sequences of this study were classified as genotype I, lineage 17 (DMV) and five as genotype I, lineage 19 (QX) (Figure 1).

Of the four DMV sequences from this study, three are identical to each other (OM525799, -800 and -801), which is foreseeable since they originated from the same US region and year. These three sequences share a nucleotide homology of 97.71% to the original DMV/1639 strain (KX529738). Sequence FLS/AZ/17 shares 89.24% nucleotide identity with the other DMV/1639-like strains from this study and 90.04% with the original DMV/1639 strain, demonstrating its uniqueness. However, OM525798 has shown high nucleotide identity (99.63–100%) to other DMV/1639-like sequences deposited in GenBank from the Delmarva peninsula isolated in 2015 (data not shown).

The five QX-like sequences from this study are more diverse since they were retrieved from different countries of Europe and Asia. The nucleotide identity between them varied from 94.14 to 99.07%, with the strains from Greece and France (OM525803 and -804) being the most closely related, and the Chinese strain (OM525802) the most distinct of the five. The nucleotide identities of these strains to the original QX (AF193423) ranged from 94.35 to 97.16%, and the strain from Slovakia (OM525806) was the only one showing nucleotide homology below 95%.

### 3.2. Multilocus Genomic Analyses

The BLAST searches targeting the top five nucleotide homologies per gene for each of the nine sequences from this study resulted in the selection of 33 DMV-like sequences and 96 QX-like sequences from GenBank, totaling 37 and 101 sequences per dataset, respectively. Eleven hits were communal to both the DMV and the QX datasets (Figure 2).

Once the QX and DMV datasets were established, sequences were aligned, and phylogenetic trees were built for each gene. We wanted to use the whole genome of the original DMV/1639/11 strain as a reference, but only the S1 gene was available on GenBank (accession number KR232396). For this reason, DMV/1639-like strains from Canada and Iowa were used for comparison (Figure 3, reference strains are highlighted in green, strains from this study are highlighted in blue). As expected, all reference sequences and our sequences clustered together with high identity in the S gene, which is used for genotyping (Figure 3, panel S). On the other hand, the distribution of strains using other genes for phylogeny is somewhat disconnected. For the most part, four groups are distributed separately: (1) the Canadian strains, (2) the Iowa strain, (3) DMV-25, -26, and -28 strains, and (4) FLS/AZ/17 (Figure 3, panels 1a, 1ab, 3a, E, M, 5a, 5b, and N). The 3b gene is relatively more consistent across the DMV dataset, with most of the strains clustering together (Figure 3, panel 3b). Interestingly, three turkey coronavirus (TCoV) sequences were selected to be part of the DMV-like dataset for having high similarity to at least one IBV gene. Although the TCoV strains are clearly outliers in the S and 5b gene phylogeny, they seem to be closely related or have a less significant distance to IBV strains in other genes. The nucleotide identities for the DMV dataset are presented in the Appendix A. Compared to all sequences retrieved from GenBank, FLS/AZ/17 showed less than 95% nucleotide identity in genes 1a and S and using the whole genome. The DMV25, -26, and -28 sequences showed less than 95% nucleotide identity to any other DMV sequence on the E gene.

For the QX dataset, the original QX strain was used as a reference (accession number MN548289) (Figure 4, highlighted in green). The QX strains from this study are highlighted in red. Within all genes, the French strain D535/4/FR/2005 was closely related to the reference QX strain (Figure 4), sharing 99.85% nucleotide identity in the whole genome. Comparably to what was observed with the DMV dataset, all QX strains clustered together using the S gene (Figure 4, panel S). A similar pattern is seen with gene 3b, with only the Hungarian strain D683/HU/06 clustering separately from the others (Figure 4, panel 3b). With the exception of strain D535/4/FR/2005 and genes S and 3b, the QX strains from this study were distributed unsystematically throughout the phylogenetic analyses of IBV genes (Figure 4). The nucleotide identities for the QX dataset are presented in Appendix A. The Greek strain d591/2/GR/05 had nucleotide identities lower than 95% in genes 1ab and in the whole genome compared to all other strains in the QX dataset. The Hungarian strain D683/HU/06 and the Slovakian strain D722/SK/06 presented with nucleotide identities below 95% in genes 1a, 1ab, 3a, 3b, N, E (Hungarian only), and whole genomes. The Chinese and French strains (D532/9/CH/05 and D535/4/FR/05) had high nucleotide identities to at least one strain from the QX dataset.

### 3.3. Prediction of Antigenic Regions

The predicted antigenic site analysis shows that the envelope protein has the highest rate of antigenic regions within the IBV genome, while accessory proteins 3a and 5b have the lowest (Table 1, Figure 5A,B).

The standard deviations of the predicted antigenic sites vary within genes and within datasets. In the DMV group, the average standard deviation was 0.62, and genes 3b, E, 3a, M and 5a showed standard deviations above the mean. The average standard deviation was higher in the QX dataset (mean SD = 0.98), with genes 3b, 3a, 5b, and 5a showing variation greater than average (Table 1).

To compare the variability of the observations within each gene, F tests were performed. The variances and the F test statistical differences between DMV and QX datasets for each gene are displayed in Figure 5C and Table 1. The greatest difference in variability between DMV and QX were in genes 3a, 3b, 5a, and 5b, with QX showing greater heterogeneity in all of them. The variances in 1a, E, and M genes were identical between DMV and QX. Even though the variances of 1ab and S genes seem graphically identical between the two datasets (Figure 5C), they are statistically different (Table 1), because the standard deviations and variances are small numbers compared to other comparisons (e.g., 3b and 3a genes).

## 4. Discussion

The IBV dynamics in the poultry population have been investigated for decades, and there are still gaps to be filled regarding the virus’ evolutionary patterns, tissue tropism, and the pathobiology of the disease caused by IBV. Here, we investigate the molecular relationships and dissimilarities between two well-known groups of IBV that have caused unusual clinical outcomes in chickens: the QX and the DMV/1639 strains and their emerging variants.

This study used four DMV/1639-like sequences and five QX-like sequences as a baseline to determine datasets based on the similarities of each of the 10 IBV genes using BLAST searches on GenBank. These isolates were obtained from collaborators that had previously reported that they belonged to IBV genotype I, lineages 17 (DMV) and 19 (QX) [12,22,24] (Figure 1). Three out of the four DMV-like isolates come from the same region in the US and are almost identical. The fourth DMV strain (FLS/AZ/17) is rather unique, being more closely related to PA/Wolgemuth/98 but with only 91% nucleotide identity on the S gene. On the other hand, all five QX strains originated from different countries in Eurasia and represent unrelated outbreaks. This difference in diversity between the baseline DMV and QX strains is the main reason why the DMV dataset (n = 37) is so much smaller than the QX dataset (n = 101) (Figure 2). In addition, the nucleotide identities between our strains and strains from GenBank were lower for DMV than for QX (Appendix A), indicating that there are few whole DMV sequences available on the database, and that those available are distantly related to the ones used in this study. QX sequences are much more abundant in GenBank than DMV sequences for two main reasons. First, QX outbreaks emerged about 10 years before the surfacing of DMV strains, allowing QX to evolve and generate more variants for longer than DMV. Second, third generation sequencing has been more largely used as a tool for the molecular surveillance of IBV in Europe and Asia than in North America, with China being the main submitter of IBV whole genome sequences in GenBank, where QX strains are widely distributed.

Although the S1 gene is rightfully consolidated as the main target for IBV classification [22], our study shows that although some strains have been classified within the same S gene classification, they seem to have arisen from completely different origins when considering other genes. For instance, DMV25, -26, and -28 cluster together with FLS/AZ/17 in genes S and 3b, but they group separately in all other genes (Figure 3). Likewise, the five QX strains do not follow a pattern of distribution in the different gene phylogenies except for S and 3b (Figure 4). There might be strains that are classified in different S1 genotypes or lineages and share almost identical genes other than the S. When investigating the pathobiology of the QX isolates used in this study, Benyeda et al. (2009) found that these five QX strains have different pathogenicity and affinity for different organs. For example, the Slovakian strain D722/SK/06 was more prone to induce ovarian lesions, while the Greek strain d591/2/GR/05 induced the mildest kidney and respiratory disease [12]. Perhaps these slight differences in tissue tropism and lesion severity are attributed to other genes, since they share the same S classification. Since recombination may occur as a result of interactions between different viruses, it is possible that these antigenic shifts rapidly shape evolutionary patterns of IBV. In the poultry industry, the use of live vaccines has a direct influence on the speed and direction of viral evolution [33]. Therefore, the different vaccines used in Eurasia and North America might have influenced the shaping and emergence of QX and DMV strains, respectively.

As previously mentioned, the 3b gene clustering follows a somewhat similar phylogenetic distribution to the S gene (Figure 3 and Figure 4), but whole genome sequences of representatives of all IBV genogroups and lineages are necessary to further evaluate this possible correlation. Studies using reverse genetics have shown that the 3b accessory protein suppresses interferon (IFN)-β production in primary cell lines [34], and that the absence of 3b leads to virus attenuation [35]. Type I interferons such as IFN-β have an important role in controlling the initial phase of the infection [36]; consequently, the lack of viral 3b protein allows for a better function of innate immunity, hindering the progression of the infection. Altogether, these findings suggest that gene 3b can be a useful tool for virus attenuation. Since the lack of accessory protein 3b does not inhibit viral replication [35], one can speculate if removing the 3b protein from vaccines could induce a more robust adaptive immune response due to a more efficient IFN-β production. It would also be interesting to see how the massive use of 3b-null viruses would shape viral evolution.

Considering all the nucleotide identities within the DMV and QX datasets, it appears that the E and N genes are potential candidates for further investigations on IBV genetic variability, since the homologies to any other sequence deposited on GenBank were below 95% in five of the nine sequences from this study. In BLAST searches using the whole genomes, one DMV (FLS/AZ/17) and three QX strains (Greece, Hungary, and Slovakia) only had hits with nucleotide identities below 95%. In all four cases, genes 1a and 1ab also showed low homologies to all sequences in the database. Gene 1ab is thought to be an important determinant of pathogenicity, bearing genomic regions that code for non-structural proteins able to attenuate viruses in vivo and in ovo [37]. It is important to remember that the 1ab gene encompasses almost 12 kbp of the IBV genome, representing almost 40% of the entire genome. In other words, the homology of whole genomes depends significantly on the homology of genes 1a and 1ab. Therefore, analyzing IBV whole genomes without investigating each IBV gene individually might be misleading and cloud conclusions about relevant genes of smaller fragment length than gene 1ab.

The EMBOSS software has been a useful tool to detect genomic regions in open reading frames, in which some amino acids associated with antigenicity are more frequent [32]. In our study, the envelope protein showed the highest rate of predicted antigenic sites in both DMV and QX datasets (Figure 5A,B). This finding suggests that the envelope protein might have a relevant role in antigenicity and immunogenicity, which could be useful since the E gene is much smaller than the S gene (E = ~300 bp and S = ~3500 bp). It has been shown that the envelope protein does not elicit high antibody titers after infection with coronaviruses [38]. Humoral responses are an important arm of immune responses and should always be a target for immune protection. However, if the E gene can stimulate some level of innate immunity that could induce cross-protection between strains, it would be a relatively easy gene to incorporate in novel vaccine technologies such as recombinant vaccines or vaccines that use mRNA technology.

Surprisingly, the S gene/spike protein did not show the highest frequency of antigenic sites among all IBV genes, even though the spike protein is the major determinant of antigenicity and tissue tropism [39]. One possible explanation for this incongruence is that the S1 portion of the S gene (~1600 bp) bears three hypervariable regions that are responsible for the most antigenic proteins of IBV [40,41]. When analyzing the entire open reading frame of the S gene, the S1 antigenic regions might have become diluted among the remaining amino acids of the spike protein.

The dispersion of predicted antigenic site datapoints within the DMV and QX datasets is noteworthy. For example, the average number of predicted antigenic sites was the highest in the E gene among all genes in both datasets, but the standard deviations and variances were also high (Table 1 and Figure 5C). This means that some strains presented with many antigenic sites, but others had less, even though they belong to the same dataset. Interestingly, despite the data distribution, the E gene variances between the QX and DMV datasets are the same, indicating that this pattern of variability within gene E might be constant within IBVs. Contrastingly, the accessory proteins 3a, 3b, 5a and 5b show significantly different variances between DMV and QX, with the QX dataset showing higher variability within these genes, whereas the variability seems to be steadier in DMV strains.

QX strains caused nephropathogenic disease when they emerged in the 1990s in Europe and Asia [7,8,9,10,11], and have subsequently been linked to reproductive disease [12,13,14]. Over time, it seems that QX strains have become more fitted to the population, becoming less pathogenic and more endemic. A similar clinical tendency is presently occurring with the DMV/1639 variants in the US and Canada [18,19,20,21], even though the conditions for viral evolution are not the same (i.e., geographic location, vaccination protocols, animal husbandry, and concomitant diseases).

Whole genome sequencing is an asset to monitor the molecular evolution of emerging IBV strains. Analyzing the entire IBV genome by individual genes provides a broader wealth of information that can be helpful in elucidating strain origins and possible pathways that lead to the selection of variants that are problematic in the field. Nevertheless, this work is essentially an in silico multitargeted assessment of the IBV genome, highlighting the variability and antigenic relevance of genes other than the S. Although the S gene is categorically the best genomic segment for classification, other genes such as the envelope and accessory protein 3b might be potential candidates to assist IBV research on antigenicity, pathogenicity, and tissue tropism.

## Figures and Tables

**Figure 1 viruses-14-01998-f001:**
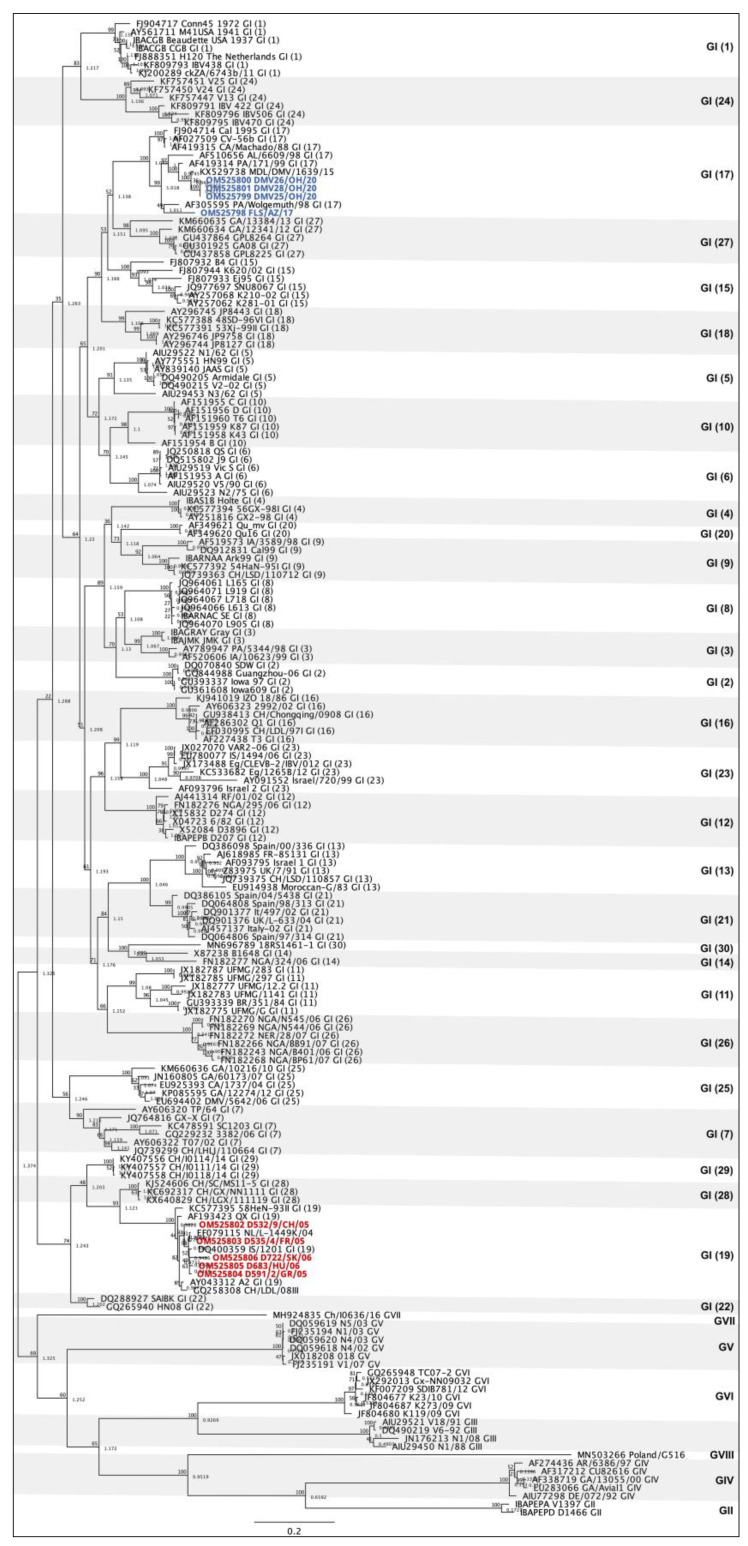
Phylogenetic tree of infectious bronchitis virus sequences based on the S1 gene. A total of 182 sequences of all 8 IBV genotypes and 37 lineages are represented. The four DMV-like sequences (genotype I, lineage 17) are highlighted in blue and the five QX-life sequences (genotype I, lineage 19) are highlighted in red.

**Figure 2 viruses-14-01998-f002:**
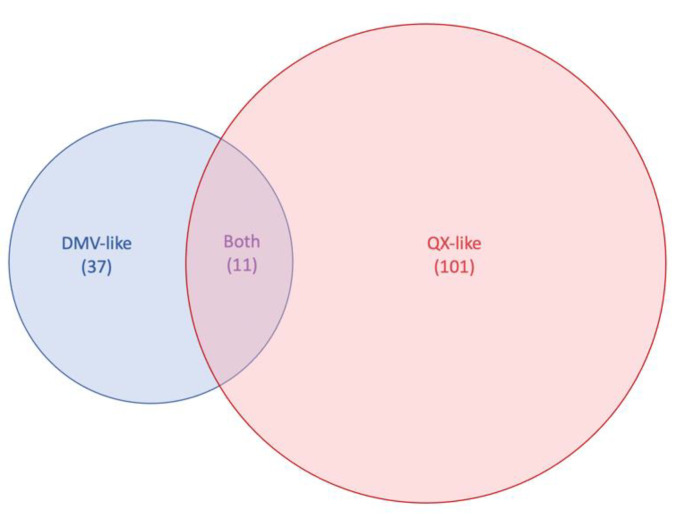
Venn diagram representing the highest BLAST hits to the DMV and QX sequences used in this this study for each infectious bronchitis virus (IBV) genes on GenBank. For each dataset, the top 5 highest nucleotide identity hits for each IBV gene was selected, totaling 37 sequences for the DMV/1639 dataset, 101 sequences for the QX dataset, and 11 sequences that were common to both.

**Figure 3 viruses-14-01998-f003:**
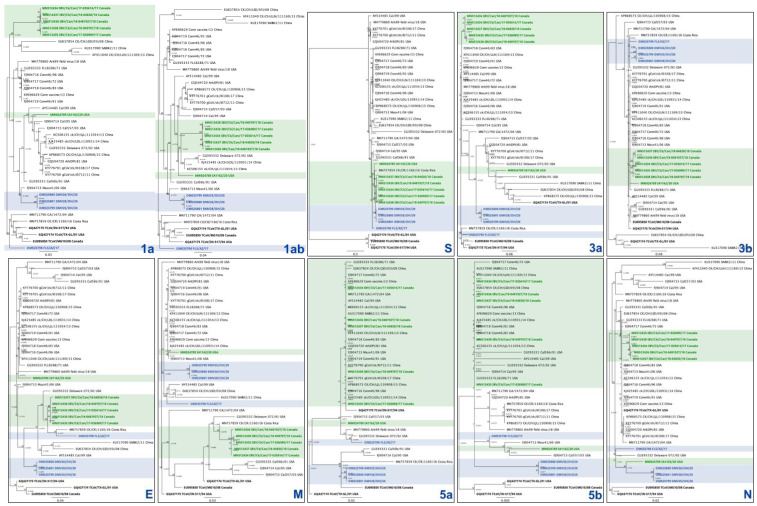
Multilocus genomic phylogenetic trees of infectious bronchitis virus sequences. A total of 37 sequences that shared high similarity to the four DMV/1639-like strains from this study on at least one gene are presented. The sequences from this study are represented in blue, while reference DMV/1639-like strains are represented in green, and TCoV are in bold. Each tree represents a different viral gene, which is labeled on the bottom right of each panel.

**Figure 4 viruses-14-01998-f004:**
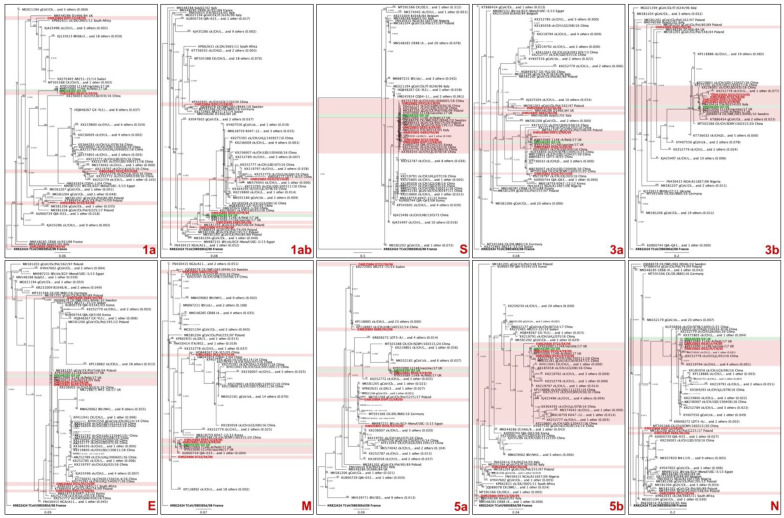
Multilocus genomic phylogenetic trees of infectious bronchitis virus sequences. A total of 101 sequences that shared high similarity to the five QX-like strains from this study on at least one gene are presented. The sequences from this study are represented in red, while the reference QX strain is represented in green. Some branches of the trees were collapsed for better visualization. Each tree represents a different viral gene, which is labeled on the bottom right of each panel.

**Figure 5 viruses-14-01998-f005:**
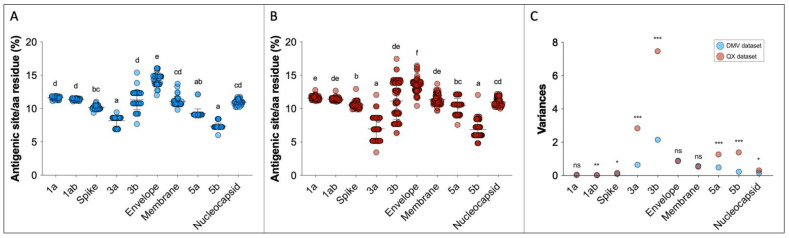
Predicted antigenic sites relative to the protein length for DMV/1639 (n = 37) (**A**) and QX (n = 101) datasets (**B**). For (**A**,**B**), superscript letters represent statistical differences (*p* < 0.05). Variances from (**A**,**B**) are represented in (**C**), and F test statistical differences between DMV and QX variances are represented as follows: ns = not significant; * *p* < 0.05; ** *p* < 0.01; *** *p* < 0.001.

**Table 1 viruses-14-01998-t001:** Average number of predicted antigenic sites relative to the protein length for DMV (n = 37) and QX (n = 101) datasets and variance comparison between QX and DMV datasets using the F test.

Protein	Average % Antigenic Sites ^1^ ± Standard Deviation		F Test Results (DMV vs. QX)		
	DMV	QX	F	F_critical_	*p* Value
Polyprotein 1a	11.61 ± 0.21	11.57 ± 0.23	1.09	1.54	0.36 ^ns^
Polyprotein 1ab	11.38 ± 0.13	11.40 ± 0.20	2.31	1.62	2.7 × 10^−3^ **
Spike (S)	10.16 ± 0.30	10.55 ± 0.40	1.74	1.62	0.03 *
Accessory 3a	8.18 ± 0.80	6.96 ± 1.69	4.43	1.63	1.5 × 10^−6^ ***
Accessory 3b	11.31 ±1.46	11.15 ± 2.73	3.48	1.62	3.13 × 10^−5^ ***
Envelope (E)	14.42 ± 0.94	13.44 ± 0.93	1.02	1.54	0.45 ^ns^
Membrane (M)	11.06 ± 0.76	11.38 ± 0.73	1.06	1.54	0.40 ^ns^
Accessory 5a	9.25 ± 0.69	10.42 ± 1.13	2.63	1.62	7.3 × 10^−4^ ***
Accessory 5b	7.36 ± 0.47	6.85 ± 1.18	6.19	1.62	1.4 × 10^−8^ ***
Nucleocapsid (N)	10.97 ± 0.40	11.00 ± 0.56	1.89	1.62	0.02 *

^1^ Number of antigenic sites predicted using EMBOSS over the total number of amino acid residues times 100. ^ns^ = not significant; * *p* < 0.05; ** *p* < 0.01; *** *p* < 0.001.

## Data Availability

The IBV whole genomes used as a basis for generating datasets used in this study are available on GenBank under accession numbers OM525798-OM5257806.

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
