# Peer review of "Infectious Bronchitis Virus: A Comprehensive Multilocus Genomic Analysis to Compare DMV/1639 and QX Strains"

_viruses, 2022, doi:10.3390/v14091998_

Round 1

Reviewer 1 Report

This manuscript (1862335) by da Silva et al. studies the molecular relationships and dissimilarities between IBV QX and DMV/1639 strains and their variants, by comparing all IBV genes independently in QX and DMV datasets. This study provides some useful information for future IBV research, especially for studies of the antigenicity, pathogenicity and tissue tropism of different QX and DMV variants and individual genes. However, due to the lack of experimental validation, the main conclusion appears not to be well unsupported.

Specific comments:

1. It would be more useful if ORF1a and 1ab regions coding for individual non-structural proteins (nsps) can be analysed separately. As some nsps may play important roles in the pathogenicity and tissue tropism of IBV, phylogenetic analysis of the dissimilarities among these nsps from different variants would be useful in future studies aiming to elucidate the functions of these proteins by other molecular biology approaches.

2. Antigenicity prediction: the prediction of E protein with high antigenicity was not supported by experiment data. In fact, in convalescent sera from COVID-19 patients and in rabbit antisera raised against purified SARS-CoV-2 virions, anti-E antibody titers are relatively low (Zhang et al., 2020, Emerging Microbes & Infections,9:1, 2653-2662). This should be discussed and caution should be taken in interpreting and discussing the prediction data.   

Author Response

REVIEWER 1

This manuscript (1862335) by da Silva et al. studies the molecular relationships and dissimilarities between IBV QX and DMV/1639 strains and their variants, by comparing all IBV genes independently in QX and DMV datasets. This study provides some useful information for future IBV research, especially for studies of the antigenicity, pathogenicity and tissue tropism of different QX and DMV variants and individual genes. However, due to the lack of experimental validation, the main conclusion appears not to be well unsupported.

We agree with the author’s comment that the conclusions from this paper are not validated by experiments. This work reflects in silico analyses and molecular work highlighting the relevance of genes other than the S gene in their contribution to antigenicity and tissue tropism. The manuscript relevance is to provide information on viral genetic and genomic patterns to support the work researchers do on IBV reverse genetics, a topic that the authors are exploring for future work. Clarification on this topic was added in lines 372-374.

Specific comments:

  1. It would be more useful if ORF1a and 1ab regions coding for individual non-structural proteins (nsps) can be analysed separately. As some nsps may play important roles in the pathogenicity and tissue tropism of IBV, phylogenetic analysis of the dissimilarities among these nsps from different variants would be useful in future studies aiming to elucidate the functions of these proteins by other molecular biology approaches.

The reviewer’s comment is valid. However, the post-translational cleavage of proteins 1a and 1ab into non-structural proteins is out of the scope of our manuscript. Likewise, we did not go into the specifics about the hypervariable regions of the S gene, for instance. In future research, targeted regions of the 1a and 1ab ORFs should be targeted for specific antigenic regions.

  1. Antigenicity prediction: the prediction of E protein with high antigenicity was not supported by experiment data. In fact, in convalescent sera from COVID-19 patients and in rabbit antisera raised against purified SARS-CoV-2 virions, anti-E antibody titers are relatively low (Zhang et al., 2020, Emerging Microbes & Infections,9:1, 2653-2662). This should be discussed and caution should be taken in interpreting and discussing the prediction data.   

We appreciate the reviewer’s comment. This publication and its repercussion were added in lines 337-340.

Reviewer 2 Report

In this study the authors use nine IBV isolates (4 of DMV/1639-like and 5 of QX-like)and 173 published sequence data to make phylogenetic inferences about the genetics difference between the two difference genotypes. In principle, I think this makes a good subject for a study that would be appropriate for this journal. However, I have some problems with the way the results and interpretation were handled. 1. First, what is the passage history of these viruses? If some of these have been grown in eggs or cell culture a few times before sequencing (M41 comes to mind), then adaptation is a potential concern. There is not much the authors can do about this, but they can give the reader the data as far as it is available. 2. Secondly, I would like to see exactly what test was done with exactly what dataset in the legend for each figure or the footnotes of each table. Even jumping back and forth between the figure and the methods section, I am still left with too many questions to be sure how to interpret the results. 3. Prior to tree inference, compute first for the best phylogenetic model (DNA substitution and site heterogeniety model) and use it during the tree inference. For each gene-based tree, include representative sequences from other major genotypes. 4. The figure is just too small - blown up to maximum size on my screen, it is still tough to make out the bootstrap numbers and names of strains. 5. The idea that S has a lower antigenic site rate than 1a、1b、3b、E、M or N is rather unexpected. I think this analysis would be better done with more than only one prediction methods. Also the results may be greatly affected by gene length, so it is better to list the number of antigenic sites of each gene. 6. Since the recombination events occurred frequently in the evolution process of IBV, is there any possibility that the differences in the phylogenetic tree branches at different gene segments may be due to the selected strains are recombinant strains? This should be discussed in the discussion section. Minor concerns 1. Line67-68: Allantoic fluid was harvested after 7 days of inoculation, which seems not the perfect time for virus harvesting of IBV, which the titer drops significantly after 48 hpi, also 18-day-old embyro eggs have only few allantoic fluid left due to embryonic development. 2. Line 149: The authors declared 8 IBV genotypes and 38 lineages, is there 38 or 37 lineages exactly? (GI-1 to 30, GII to GVIII)?

Author Response

REVIEWER 2

In this study the authors use nine IBV isolates4 of DMV/1639-like and 5 of QX-like and 173 published sequence data to make phylogenetic inferences about the genetics difference between the two difference genotypes. In principle, I think this makes a good subject for a study that would be appropriate for this journal. However, I have some problems with the way the results and interpretation were handled.

  1. First, what is the passage history of these viruses? If some of these have been grown in eggs or cell culture a few times before sequencing (M41 comes to mind), then adaptation is a potential concern. There is not much the authors can do about this, but they can give the reader the data as far as it is available.

The authors agree with the reviewer’s comment. The in vitro handling of viruses is a relevant bias that affects host adaptability and consequently predominant virus subpopulations. However, we chose to do virus isolation to be able to perform live bird experiments with these viruses in the future. All viruses were isolated in the first inoculation/passage into embryos (added in lines 69-71).

  1. Secondly, I would like to see exactly what test was done with exactly what dataset in the legend for each figure or the footnotes of each table. Even jumping back and forth between the figure and the methods section, I am still left with too many questions to be sure how to interpret the results.

Information about the datasets was added to the legend of figure 2 (lines 180-182). Supplemental tables were added to provide the raw data about the antigenic site prediction analyses (Tables S3 and S4, lines 380-384).

  1. Prior to tree inference, compute first for the best phylogenetic model (DNA substitution and site heterogeniety model) and use it during the tree inference. For each gene-based tree, include representative sequences from other major genotypes.

We appreciate the reviewer’s comment. We did not add representative sequences from all genotypes because there aren’t enough available sequences for genes other than the S in the database. We chose to do database-specific phylogenetic trees (DMV and QX databases in Figures 3 and 4, respectively) since very few genotypes would be appropriately represented otherwise.

  1. The figure is just too small - blown up to maximum size on my screen, it is still tough to make out the bootstrap numbers and names of strains.

We apologize for the figure size. We tried to upload the best quality figure possible for zooming in. I don’t think enlarging the bootstrapping numbers would make it easier since they would cover branches and sequence labels. We can discuss better options with the copy editor should the paper gets accepted.

  1. The idea that S has a lower antigenic site rate than 1a1b3bEM or N is rather unexpected. I think this analysis would be better done with more than only one prediction methods. Also the results may be greatly affected by gene length, so it is better to list the number of antigenic sites of each gene.

The antigenic site rate per gene accounted for the gene length as described in lines 130-132. Supplemental tables with the raw data were added for clarification (lines 132, 380-384).

  1. Since the recombination events occurred frequently in the evolution process of IBV, is there any possibility that the differences in the phylogenetic tree branches at different gene segments may be due to the selected strains are recombinant strains? This should be discussed in the discussion section.

Yes, it is possible that recombination is happening within different IBV genes when more than one predominant subpopulation is present. This was mentioned in lines 299-301.

Minor concerns

  1. Line 67-68: Allantoic fluid was harvested after 7 days of inoculation, which seems not the perfect time for virus harvesting of IBV, which the titer drops significantly after 48 hpi, also 18-day-old embyro eggs have only few allantoic fluid left due to embryonic development.

The reviewer is correct. The embryos are incubated for up to 7 days post-inoculation, but the vast majority of them died between 3 and 4 days post-inoculation. This was fixed in line 67.

  1. Line 149: The authors declared 8 IBV genotypes and 38 lineages, is there 38 or 37 lineages exactly? (GI-1 to 30, GII to GVIII)?

The reviewer is right, there are 37 lineages. Thirty lineages belonging to genotype I and one lineage of each genotype II-VIII. This was fixed in lines 107 and 154.

Round 2

Reviewer 2 Report

The author has revised and answered some of the review comments, but I am still concern about the final result presented now.

First,all the ML-trees were constructed using the same nucleotide substitution model :GTRGAMMAI. However, the best fit models of each gene should be tested and used in the analysis. Eventhough I can’t see clearly the bootstrap value, some of them seems quite low and not acceptable for the tree construction.

Second, only one method was used for antigenic site prediction, the authors should compare the antigenic sites predicted by different software or on-line method to make the results more reliable. 

The supplementary table S3 and S4 are missing in my system.  supplementary file(s)

Author Response

The author has revised and answered some of the review comments, but I am still concern about the final result presented now.

Firstall the ML-trees were constructed using the same nucleotide substitution modelGTRGAMMAI. However, the best fit models of each gene should be tested and used in the analysis. Eventhough I can’t see clearly the bootstrap valuesome of them seems quite low and not acceptable for the tree construction.

We agree with the reviewer that it would be valuable to test the best fit models for each gene of each dataset. We understand that all models are to some level incorrect since they are designed to simplify reality. Nevertheless, the GTR model accounts for nucleotide substitution frequency and rate parameters, being trustworthy and widely used by several research groups all over the world. Considering the size of our datasets and that we are using 10 different IBV genomic open reading frames for our analysis, the GTR model is an appropriate and reliable probabilistic method to account for all 20 phylogenetic trees in Figures 3 and 4.

We agree that the bootstrap values are low on some occasions; low bootstrap values simply indicate lack of branch support, but we are not determining any classification method. The aim of our study and Figures 3 and 4 is to demonstrate the phylogenetic distribution and evolutionary relatedness of IBV sequences that belong to the same S1 genotype. We are not setting up any parameters for genotyping.

Second, only one method was used for antigenic site prediction, the authors should compare the antigenic sites predicted by different software or on-line method to make the results more reliable. 

We respectfully disagree with the reviewer’s suggestion. There is no reason to think the method used in our paper is unfit to predict antigenic sites. The EMBOSS software simply counts the number of cysteine-, leucine-, and valine-rich regions within each gene. This methodology has been reliable in predicting antigenic sites. Redoing the analysis using a different tool will involve many hours of work and most likely will provide similar results.

The supplementary table S3 and S4 are missing in my system. supplementary file(s).

We apologize for the mistake. The Supplemental Tables S3 and S4 were uploaded this time.
